# Integrated group antenatal and pediatric care in Haiti: A comprehensive care accompaniment model

**Meredith Casella Jean-Baptiste**[1]*, **Marc Julmisse**[2], **Oluwatosin O. Adeyemo**[3],
**Thamar Monide Vital Julmiste**[4], **Jessica L. Illuzzi**[3]

1 Maternity Department, Hôpital Universitaire de Mirebalais, Zanmi Lasante/ Partners In Health, Mirebalais, Haiti, 2 Executive Direction, Zanmi Lasante/ Partners In Health, Mirebalais, Haiti, 3 Department of Obstetrics, Gynecology, and Reproductive Sciences, Yale School of Medicine, New Haven, Connecticut, United States of America, 4 Journey 9/ Nursing Center of Excellence, Zanmi Lasante/ Partners In Health, Mirebalais, Haiti

☯ These authors contributed equally to this work.
* mjeanbaptiste@pih.org

**Data Availability Statement:** All relevant data are within the manuscript and its Supporting Information files.

## Abstract

### Introduction

The J9 Plus (J9) maternal-child accompaniment program is based on four pillars: group antenatal care (GANC), group pediatric care, psychosocial support, and community-based care. We aimed to evaluate the impact of the J9 model of care on perinatal outcomes.

### Methodology

We conducted a convergent mixed methods study of maternal-newborn dyads born in 2019 at Hôpital Universitaire de Mirebalais. Quantitative data was collected retrospectively to compare dyads receiving J9 care to usual care. A secondary analysis of qualitative data described patient perspectives of J9 care.

### Results

Antenatal care attendance was significantly higher among women in J9 (n = 524) compared to usual care (n = 523), with 490(93%) and 189(36%) having >4 visits, respectively; p <0.001, as was post-partum visit attendance [271(52%) compared to 84(16%), p<0.001] and use of post-partum family planning methods [98(19%) compared to 47(9%), p = 0.003]. Incidence of pre-eclampsia with severe features was significantly lower in the J9 group [44 (9%)] compared to the usual care group [73(14%)], p <0.001. Maternal and neonatal mortality and low birth weight did not differ across groups. Cesarean delivery [103(20%) and 82 (16%), p<0.001] and preterm birth [118 (24%)] and 80 (17%), p <0.001] were higher in the J9 group compared to usual care, respectively. In the qualitative analysis, ease of access to high-quality care, meaningful social support, and maternal empowerment through education were identified as key contributors to these outcomes.

**Funding:** Produced with support from the W.K. Kellogg Foundation. The funders had no role in study design, data collection and analysis, decision to publish, or preparation of the manuscript.

**Competing interests:** The authors have declared that no competing interests exist.

## Conclusion

Compared to usual care, the J9 Plus maternal-child accompaniment model of care is associated with increased engagement in antenatal and postpartum care, increased utilization of post-partum family planning, and lower incidence of pre-eclampsia with severe features, which remains a leading cause of maternal mortality in Haiti. The J9 accompaniment approach to care is an empowering model that has the potential to be replicated in similar settings to improve quality of care and outcomes globally.

## Introduction

Improving access to and engagement with quality healthcare during pregnancy, birth, and the months after giving birth has the potential to identify mothers and babies at risk and assist them in achieving healthy outcomes. Group antenatal care (GANC) is a model of care that has been shown to increase attendance at antenatal healthcare visits and improve outcomes in resource-limited regions in many countries, including Nigeria, Kenya, Ghana, Iran, Senegal, Malawi, Tanzania, Nepal, and Rwanda [1–9]. GANC has been linked to decreases in preterm birth rates [10], increased adherence to antenatal care visits [1, 2, 11], use of family planning [3], exclusive breast-feeding [3], and improved patient and provider satisfaction [4–8].

GANC allows providers to spend more time on rich content that is driven by needs of individuals within the group, allows participants to develop a social support network throughout their pregnancy and postpartum periods, and reduces wait time for care [12]. Group pediatric visits offer the benefit of newborn screenings and vaccinations, increased time for education, and parent to parent support. Group visits have been found to be equivalent to individual well-child care [9, 13, 14] and offer the possibility for such care in regions without routine well-child care.

In Haiti, the maternal mortality rate is the highest in the Western Hemisphere at 529 per 100,000 live births [15], with a lifetime risk of maternal death for 1 in 80 women [16]. Similarly, the neonatal and infant mortality rates in Haiti are extremely high with 24 deaths per 1000 live births in the first 28 days of life [17] and 45 deaths per 1000 live births under the age of 1 year [18, 19]. According to the Haitian Ministry of Health (MSPP), 78% of Haitians lack access to health care due to financial or geographic barriers [20]. Per capita income is less than two dollars per day making mothers and infants in rural areas particularly vulnerable to disease and needless death [21]. Pregnant women access antenatal care sporadically and usually for emergent problems only, with only 42% giving birth in health care facilities with a skilled birth attendant [16]; postpartum and well-baby visits are not a routine part of the health care system.

Partners in Health/ Zanmi Lasante (PIH/ZL) aimed to increase access to quality services as well as enhance interventions to reduce maternal and newborn mortality and morbidity in the Central Plateau at Hôpital Universitaire de Mirebalais (HUM). In July 2018, a collaborative, interdisciplinary, accompaniment approach to care for pregnant women and infants up through and beyond nine months of life, referred to as the "Journey to 9 Plus" (J9) began. Drawing on the strengths of GANC demonstrated in other countries, J9 facilitates access to care and provides a holistic integrated approach based on the accompaniment model that is the foundation of Partners In Health (PIH) work. The accompaniment model is characterized by excellent clinical care with synchronous social support [22]; it is both a philosophy as well as a rubric for programmatic design [22]. Based on pragmatic solidarity with the poor, it

Traditional antenatal and pediatric care in Haiti is often characterized by the absence of set appointment times for patients and providers leading to long wait times. Patients may still not be seen by a provider, necessitating the patient return on another day to be seen or to receive lab results ultimately a missed opportunity for care provision.

**Group Antenatal Care (GANC)**: Group antenatal and pediatric care in the J9 Plus model provides guaranteed appointment times for patients. Visits are conducted in a group setting, with care provided by nurse midwives and nurses, and high-risk patients enrolled in parallel external consultations with OB/GYNs. Participation in J9 Plus is offered to patients who are beginning their second trimester of pregnancy (16-20 weeks gestation); have already had a first antenatal visit; are residing in the Mirebalais catchment area; and plan to deliver in Haiti.

**Group Pediatric Care**: Patients who participate in group antenatal care (GANC) are transitioned to group pediatric care along with their neonates following delivery. GANC visits are monthly in the beginning of pregnancy then transition to bi-monthly starting at 28 weeks and weekly at 36 weeks. Group pediatric care is monthly until the newborn is 3 months of age, then every trimester (3-months) through the first year.

**Psychosocial support:** J9 Plus program maternal and pediatric teams in collaboration with the psychologist on the mental health team developed the post-traumatic stress disorder (PTSD) and depression screening tool for maternal child health, for which every woman is screened for PTSD and depression at her first group visit and during the immediate postpartum period.

**Community-based care:** Community health workers alongside the J9 Plus clinical and psychosocial team, which is comprised of psychologists and social workers, visit patients at home to provide follow up care and further assess social determinants of health such as access to clean water, sewage disposal and distance between home and the hospital amongst other factors. During these integrated in-home assessments, inclusive maternal-newborn dyad clinical evaluation and teaching allows for access to home-based family planning. Multi-generational community teaching on key J9 messages exposes families to the same information that mothers receive in J9, and psychosocial counseling of mother and partners at home permits unique access while normalizing mental health support.

**Fig 1. Journey 9 plus programmatic pillars.**

proposes building long-term relationships and walking with the patient [23], rather than leading [22].

The four main pillars of the J9 Plus model of care are: 1) group antenatal care (GANC); 2) group pediatric care; 3) psychosocial evaluation and support; and 4) community-based care with a home visit and assessment [see Fig 1]. Interdisciplinary referrals are made as needed throughout the patient's participation in the program. The goal of this approach is to increase maternal and infant survival through education, supportive networks of peers and providers, assessment of and aid in mitigating social determinants and barriers to health, and earlier identification and referral of threats to maternal and neonatal health.

## Objectives

The primary goal of this study was to evaluate the maternal and neonatal health outcomes of the comprehensive J9 Plus program implemented in the Central Plateau of Haiti and to explore the components and characteristics of the program that may be contributing to its effectiveness as an intervention.

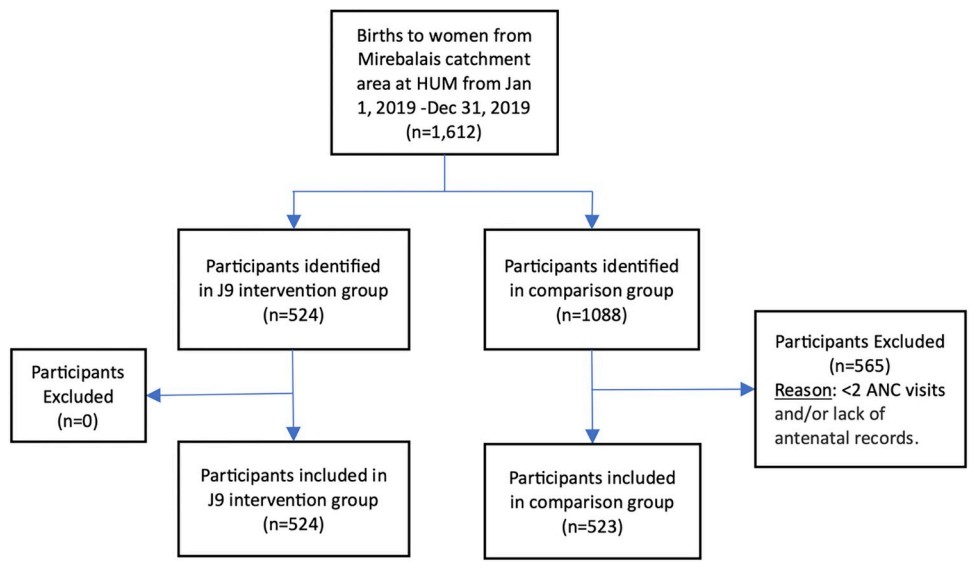

**Fig 2. Inclusion criteria of women delivering in 2019 at HUM.**

## Methodology

A convergent mixed methods retrospective cohort study was conducted to evaluate the impact of the J9 Plus program in the Mirebalais catchment area at HUM. For a full description of the J9 Plus program including enrollment details, see Fig 1.

### Ethics

This study involved human participants and was approved by the Zanmi Lasante Institutional Review Board (reference #: ZLIRB03022021). The IRB granted a waiver of informed consent for this retrospective analysis of deidentified clinical data. Retrospective quantitative data was accessed, de-identified, and stored for analysis on a secure platform on 01/10/2021. A select group of participants gave informed consent for their focus group transcripts to be used for qualitative analysis. These transcripts were accessed and translated on 22/01/2022 and then stored in a de-identified manner on a secure platform for analysis.

### Quantitative data

The intervention group was composed of patients who were enrolled in the J9 Plus program and gave birth at HUM who resided in the defined Mirebalais catchment area between January 1 and December 31, 20219. The comparison/ usual care group was composed of all patients who also gave birth at HUM between the same period as the intervention group at HUM and who resided in the same defined Mirebalais catchment area as the intervention group but were not enrolled in J9 Plus. Patients who did not live in the defined Mirebalais catchment area, for whom we did not have antenatal records available, or who had <2 antenatal visits at HUM (and therefore may have had antenatal care elsewhere) were excluded.

Perinatal outcomes of interest included maternal and neonatal mortality and morbidities, such as preeclampsia with severe features, hemorrhage, preterm birth, and low birth weight. Other outcomes of interest included mode of delivery, Apgar scores, number of antenatal and postpartum visits, and family planning usage.

Data was extracted through chart review including electronic medical records (EMR), paper charts, and hospital and J9 program registers. Maternal variables included chart number,

age, estimated date of delivery, last menstrual period, gravity, parity, number of antenatal visits, number of postpartum visits, postpartum family planning visit and method, mode of delivery (vaginal, cesarean, or instrumental vaginal delivery), and presence or absence of antepartum, intrapartum, and postpartum complications such as preeclampsia, eclampsia, placental abruption, postpartum hemorrhage, fetal demise, and maternal mortality. Neonatal variables included chart number, date of birth, gestational age at birth (weeks and days), Apgar score at 5 minutes (and 10-minute Apgar scores when applicable), birthweight, neonatal mortality, and number of days of hospitalization, if admitted.

Maternal variables extracted from the EMR included date of birth and family planning visits (date and method type). All remaining maternal and neonatal variables were extracted from paper charts. All data from paper charts were entered into a CommCare data collection platform.

Data was de-identified and kept confidential and anonymous during data analysis. Quantitative data was cleaned and validated after export (EMR) or data collection (CommCare App). Characteristics and outcomes of patients in the cohort enrolled in J9 were compared to those who received usual care using STATA-18 and confirmed by a statistician. Continuous variables were compared using t-tests or linear regressions and categorical variables were compared using chi-square or logistic regression. Odds ratios and confidence intervals were calculated when appropriate.

## Qualitative data

We conducted secondary analysis of pre-existing qualitative data collected as part of a J9 program evaluation in order to learn about the experiences of participants in the J9 Plus program during antenatal care, birth, and postpartum, including the components and characteristics of the program that participants identified that may be contributing to its effectiveness as an intervention

Between March 10–13, 2020, J9 participants, who had given birth between January 1, 2019 and December 31, 2019 and were engaged in J9 group pediatric care at HUM, were invited to participate in one of three focus group discussions (FGD) after their group visits as part of a planned J9 program evaluation. The FGDs were conducted in Haitian Creole by a Haitian sociologist who was not a member of the J9 clinical team. Participants were asked to describe their experiences of antenatal and post-partum care, home-visits, and psychosocial support received as part of J9 Plus. The sessions were recorded with verbal patient consent and later transcribed into Haitian Creole for J9 programmatic evaluation. Consent to review and analyze responses for research purposes was subsequently obtained from patients. Patients whose consent for qualitative analysis of the responses could not be obtained were excluded from this study. As a result, only one FGD which consisted of eight (8) participants was included for qualitative analysis in this study. The FGD transcripts were translated verbatim from Haitian Creole to English by a bilingual speaker for review and analysis.

Prior to analysis, the transcription was quality-checked by a study team members (MCJB and TMVJ), who are both fluent in English and Creole. They reviewed the entire transcript both in English and in Haitian Creole and compared it against the audio recording. Any differences in transcription were highlighted for further review and discussed prior to finalizing the transcript for coding.

The group Interviews were analyzed using thematic analysis and inductive techniques from Grounded Theory to develop themes [24]. MCJB and OOA reviewed the transcripts and independently performed In Vivo coding. MCJB and OOA discussed both code application and themes and agreed on the final code book. They wrote reflexive memos as part of this process

and developed emerging concepts and themes which were refined. Integrative memos were noted, and all results were discussed with other team members.

## Results

### Quantitative results

During the study period, 1,612 women from the Mirebalais catchment area gave birth at HUM. After exclusions for <2 antenatal visits and lack of antenatal records, 524 women in the J9 intervention group and 523 women in the usual care group met inclusion criteria for comparison [See Fig 2].

Table 1 shows the baseline characteristics of the two groups. The mean maternal age was 29 years for the J9 group and 28 years for the usual care group. Maternal age was normally distributed however, there was a lower proportion of teenagers in the J9 group; $p = 0.001$. Parity was similar between the two groups; specifically, 161 (31%) vs 143 (32%) of the respective groups were nulliparous, 340 (66%) vs 286 (63%) were multiparous, and 16 (3%) vs 18 (4%) were grand multiparous ($\geq 5$ deliveries); $p = 0.683$.

Given the significant difference in maternal ages between the J9 and usual care groups, we controlled for maternal age in subsequent analysis. P-values were calculated using linear regressions for all continuous variables, and logistic regressions conducted for all binary variables, while controlling for maternal age in both cases.

Maternal outcomes are detailed in Table 2. Numbers of antenatal care visits differed significantly among the two groups, with 34 (7%) versus 334 (64%) having less than 4 visits, 367 (70%) versus 179 (34%) having 4 to 7 visits, and 123 (23%) versus 10 (2%) having more than 7 visits among women in the J9 group compared to those receiving usual care, respectively; p-<0.001. Prevalence of sexually transmitted infections (STI) in pregnancy did not differ between the two groups. Pre-eclampsia with severe features was diagnosed less frequently in the J9 group compared to the usual care group: 44 (9%) versus 73 (14%), respectively; p-value<0.001.

Cesarean delivery occurred more frequently among the J9 participants compared to those in usual care [103 (20%) versus 82 (16%)] as did instrumental vaginal delivery [25 (5%) versus 1 (0.2%); p-value<0.001], respectively. Post-partum hemorrhage did not differ significantly between the two groups [4 (1%) versus 8 (2%); p-value = 0.250] nor did maternal mortality [3 (0.6%) versus 2 (0.4%); p-value = 0.622], respectively.

**Table 1. Baseline characteristics of J9 and usual care groups.**

| BASELINE CHARACTERISTICS | J9 Group n (%) | Usual Care n (%) | p-value |
|---|---|---|---|
| **Age** | **n = 524** | **n = 523** | |
| 12–19 years | 25 (4.8) | 58 (11.1) | <0.001 |
| 20–34 years | 393 (75.0) | 370 (70.8) | |
| 35 years and older | 106 (20.2) | 95 (18.2) | |
| Mean (Standard Deviation) | 29.3 (6.17) | 27.9 (6.65) | <0.001[††] |
| **Parity** | **n = 517[†]** | **n = 447[†]** | |
| Nulliparous | 161 (31.1) | 143 (32.0) | 0.683 |
| Multiparous | 340 (65.8) | 286 (64.0) | |
| Grand Multiparous ($\geq$ 5 prior deliveries) | 16 (3.1) | 18 (4.0) | |

[†] Note: The n changed for certain variables (where mentioned) due to missing data

[††] T-test conducted for this p-value

**Table 2. Maternal outcomes for J9 and usual care groups.**

| MATERNAL OUTCOMES | J9 Group n = 524[†] n (%) | Usual Care n = 523[†] n (%) | OR [95% CI] [††] | p-value [††] |
|---|---|---|---|---|
| **Antenatal visit attendance** | **n = 524** | **n = 523** | | |
| 0 to 3 visits | 34 (6.5) | 334 (63.9) | | <0.001 |
| 4 to 7 visits | 367 (70.0) | 179 (34.2) | | |
| 8 or more visits | 123 (23.5) | 10 (1.9) | | |
| **STI [†††] in Pregnancy** | **n = 505** | **n = 518** | | |
| | 11 (2.2) | 22 (4.2) | 0.48 [0.23, 1.01] | 0.052 |
| **Placental Abruption** | **n = 497** | **n = 521** | | |
| | 0 (0) | 2 (0.4) | – | –[omitted] |
| **Pre-eclampsia with severe features/ Eclampsia** | **n = 506** | **n = 521** | | |
| | 44 (8.7) | 73 (14.0) | 0.56 [0.37, 0.83] | <0.001 |
| **Delivery Type** | **n = 524** | **n = 523** | | |
| Caesarian | 103 (19.7) | 82 (15.7) | | <0.001 |
| Vaginal | 396 (75.6) | 440 (84.1) | | |
| Instrumental | 25(4.8) | 1 (0.2) | | |
| **Post-Partum Hemorrhage[††††]** | **n = 501** | **n = 522** | | |
| | 4 (0.8) | 8(1.5) | 0.49 [0.15, 1.65] | 0.250 |
| **Maternal Mortality** | **n = 524** | **n = 523** | | |
| | 3 (0.6) | 2 (0.4) | 1.6 [0.26, 9.60] | 0.622 |
| **Post-Partum Visit ≤ 6 weeks** | **n = 524** | **n = 523** | | |
| | 271 (51.7) | 84 (16.1) | 5.47 [4.09, 7.32] | <0.001 |
| **Family Planning** | **n = 524** | **n = 523** | | |
| Number of patients starting a method within 12 months after birth | 98 (18.7) | 47 (9.0) | 2.52 [1.73, 3.67] | <0.001 |

[†] Note: The n changed for certain variables (where mentioned) due to missing data

[††] The OR, 95% CI, and p-values were calculated using a logistic or linear regression where applicable, controlling for maternal age

[†††] Sexually Transmitted Infections (STI) diagnosed in pregnancy

[††††] Post-partum hemorrhage as recorded in the maternal medical record as a complication (yes/no)

Two-hundred-seventy-one (52%) of J9 participants versus 84 (16%) of women in usual care attended their post-partum visit appointment; p-value<0.001. Utilization of a family planning method was more common among J9 participants than among those in usual care [98 (19%)] versus 47 (9%); p-value<0.001], respectively.

Neonatal outcomes are detailed in Table 3. Rates of low birth weight were similar among the J9 group and those in usual care [78 (17%) compared with 83 (16%), respectively; p = 0.741]. Among the Apgar categories, more babies in the J9 group were born with Apgar scores of 4–7 compared to those in usual care [102 (22%) versus 77 (15%)], respectively, coinciding with a decrease in Apgar scores of 8 and greater among J9 participants; p = 0.003.

Higher incidences of preterm delivery were found in the J9 group compared to the usual care group, most notably between 28 weeks and 36 weeks and 6 days of gestation for 117 (24%) versus 78 (17%) of all births in each group, respectively; p = 0.019. Mean gestational age among J9 neonates (37.7 weeks) was lower than neonates born to mothers who received usual care (38.3 weeks); p<0.001.

There was no difference in NICU admissions between the J9 and usual care group [31 (6.1%) and 28 (6.2%), respectively; p = 0.800], nor were there differences in neonatal mortality, with 6 (1.2% and 1.3%) in the groups; p = 0.833.

**Table 3. Newborn outcomes for J9 and usual care groups.**

| NEWBORN OUTCOMES | J9 group n = 528[†]<br>n (%) | Usual care<br>n = 532[†]<br>n (%) | OR [95% CI] [††] | p-value [††] |
|---|---|---|---|---|
| **Low Birth Weight** | **n = 466** | **n = 514** | | |
| <2500g | 78 (16.7) | 83 (16.1) | 1.06 [0.75, 1.49] | 0.741 |
| **Apgar Score [†††]** | **n = 464** | **n = 521** | | |
| Apgar 0 (IUFD) | 5 (1.1) | 7 (1.3) | – | 0.700 |
| Apgar 1–3 | 3 (0.7) | 0(0) | | 0.003 |
| Apgar 4–7 | 102 (22.0) | 77 (14.58) | | |
| Apgar ≥ 8 | 354 (76.3) | 437 (84.5) | | |
| **Gestational Age** | **n = 493** | **n = 470** | | |
| <28 weeks | 1 (0.2) | 2 (0.4) | | 0.019 |
| 28–36+6 weeks | 117 (23.7) | 78 (16.6) | | |
| ≥37 weeks | 375 (76.1) | 390 (83.0) | | |
| Mean (Standard Deviation) | 37.7 (2.34) | 38.4 (2.34) | | <0.001[††††] |
| **NICU [†††††] Admissions** | **n = 509** | **n = 456** | | |
| | 31 (6.1) | 28 (6.2) | 0.93 [0.55, 1.59] | 0.800 |
| **Neonatal Demise** | **n = 478** | **n = 485** | | |
| | 6 (1.2) | 6 (1.3) | 0.88 [0.28, 2.77] | 0.833 |

[†] Note: The n changed for certain variables (where listed) due to missing data

[†] Nine (9) sets of twins in the usual care group and four (4) sets in the J9 cohort explains the newborn group size differences as compared to maternal cohorts.

[††] The OR, 95% CI, and p-values were calculated using a logistic or linear regression where applicable, controlling for maternal age

[†††] Apgar score at 5 and 10 minutes were collected and the highest score selected.

[††††] T-test conducted for this p-value

[†††††] NICU (Neonatal Intensive Care Unit)

## Qualitative results

Transcribed FGD data was coded and analyzed for themes and sub-themes [see Table 4]. Four main themes emerged from the data: 1) Empowerment through provision of education and information; 2) Efficiency and Integrated care; 3) Social support or "The love that is in J9"; and 4) Perception of reduced morbidity and mortality.

### Theme 1: Empowerment through provision of education and information

*"...With the education it was very useful, we have a healthy baby and we saved our lives"*

A recurring sentiment from participants was how the education and information they received from J9 program made them feel empowered. We developed four subthemes that capture the information participants found important.

*Subtheme*: *Warning signs in pregnancy*. Participants expressed feelings of empowerment because they were educated about warning signs in pregnancy which led them to advocate for themselves in times of emergencies and make self-referrals to care when they had concerning symptoms. One patient explains how education facilitated the dispelling of myths around bleeding: "My mother told me that it was good, not to worry, it is my body preparing myself..." as well as the water breaking from "drinking too much water." This participant self-referred to the hospital for care and she was able to advocate for herself; knowing the warning

**Table 4. Themes and subthemes with illustrative quotations.**

| Themes and Subthemes with Illustrative Quotations | |
|---|---|
| Theme or Subtheme | Illustrative Quotation |
| *Theme 1: EMPOWERMENT through provision of education and information* | |
| Warning signs in pregnancy | "Thanks to the education they gave us, one of the warning signs; one thing that was useful for me and helped me, but I can say that they helped me the most were the warning signs. Well thanks to the trainings, I was at home one day when I saw that I started to have a streak of blood. And my mother told me that it was good, not to worry, it is my body preparing myself, I am getting ready. But the fact that they taught me that it was a warning sign, immediately, I put clothes on and I came and when I arrived at the hospital, I went to Triage (Women's Health ER) and when they checked me and they told me I am not yet ready because I was in latent labor, when they sent me outside and then I saw that I was bleeding and broke my water. There was someone who told me that "Oh, you drank too much water", but me, the fact that J9 educated us, I saw that it was a warning sign and when I went back inside and told them that I broke my water, immediately they put me. . . they gave me an IV and gave me a bed. Thanks to the training that they did for us, when I was in latent labor, when I broke my water in latent labor. So, the fact that I had the education J9 gave me, then I have my baby. But if I stayed when I broke my water, amniotic fluid, there wasn't any inside and I could have lost the baby. Then with the education it was very useful, we have a healthy baby and we saved our lives." |
| Nutrition and Lifestyle changes | "Our experience in J9 has improved our health and our children's health in the sense that the training and education they did for us; they talked to us about the dietary food groups that we need to eat, the different foods that we need to consume (the body builders, the protectors and others) then when we eat these foods, it helps us have a healthy baby and us too, we are in good health, even after we deliver. " |
| Information on care of the baby | ". . . For the baby always, they always talked with us, they always told us . . . to be sure. be sure to watch over the babies, to always be mindful of how we take care of the babies." |
| Empowerment/ teaching others | ". . . I have been doing this because I already gave many others. I explained to them the same way I have been taught during the training. I have done a lot of trainings with other people; I have already referred two people to whom I explained that. To date, I think I will continue to let people know what is in J9, because I love J9 so much, thanks." |
| *Theme 2: Efficiency of the program* | |
| Reliable and efficient antenatal care | "No, I did not have experience anywhere else, because usually when you go for a prenatal visit, you spend almost the whole day, to get your chart, then they send you to get lab tests, sometimes you aren't even able to get seen. But with j9, it is different because there is a group and they call you sometimes, they call you for your appointment and when you come, they have you sit down and while you are learning they give you your prenatal check and you don't even take a lot of time. If it is 8 o'clock you have an appointment, by 10 o'clock or before 10 o'clock you have left and are heading home. So as a pregnant woman sometimes you are not feeling well (have discomfort) so with the J9 program, it has everything you need; it is all that I need. " |

*(Continued)*

**Table 4.** (Continued)

| | |
|---|---|
| Integrated care: Connection between antenatal care and hospital care | "For me, what I loved about the prenatal consultations while I was pregnant was that sometimes I had discomforts and I did not know what was going on if I wasn't feeling well. It was when I arrived that they told me and then they did my checkup and then when. . . baby . . . the baby's heart was beating very fast and I never knew it and when I arrived in J9, they told me and they sent me to the Women's Emergency Room and there they measured me and checked everything and they sent me back to J9 again. That I loved. I don't have anything bad to say about that." |
| **_Theme 3: "The love that is in J9"_** | |
| Personable and respectful care from providers | ". . .I enrolled in J9 and I have found a lot of great things in J9, even though it's my fourth child, I have found that they really take care of us. When I come in for a consultation they call me for my checkup, and they always welcome us very well and after I delivered as well, you always receive a warm welcome. That makes me happy for J9 and I always will be. I will never forget J9 for the way they took care of us." |
| Support from other patients | "So, what impressed me the most. . .. It's true that the training and education they. . . they gave us was really important, it really help me. But the love that is inside J9, that impressed me even more. Because the fact that they brought us together in a group, it was like we were in a family. It was like we are sisters. When we delivered, we asked after one another, and we were like we loved delivering and when I gave birth, I remember that I found 2 other mothers, after that there was a third and then there was a fourth. That was like there were four sisters that were there, it was like we were shoulders for one another and when we delivered, we watched over the beds. When it was time to leave, they said I saw you deliver, I will wait for you. And then it was like. . . just after that we talked on the phone, one will ask after the others, how are the babies, how is the baby. It is like we came to be a family. So, the love that is inside J9, I love that." |
| **_Theme 4: Perception of reduced morbidity and mortality_** | |
| Prevention for mother | "Thanks to the J9 program everyone can notice that there are fewer children dying and fewer women who die from eclampsia as well. Because there is post-partum eclampsia, I could have had that because when they called me to come for the postpartum visit, I never knew that my blood pressure was so high. I had a blood pressure of 17 over 15 (170/150) if I am not mistaken. So, when I came and they told me my blood pressure was that high, if I did not. . . if they did not call me for my post-partum visit, I could have had eclampsia or I could have died. So, I need to congratulate and thank the J9 program. . ." |
| Prevention for infant | "Well, the strong points that I liked about it, is that when they came to visit me, they saw the baby and they noticed that the umbilical cord had not fallen off yet. Me myself, I had not noticed that and when they examined the baby, they saw that the umbilical area was not normal, and they said to me to urgently go to the hospital. The fact that it was late, and they told me to go very early the next morning to the hospital. Then when I came to the J9 pediatrician with her, then it was a really good thing. The fact that if they had not come and perhaps she could have had an infection or even something worse (could have happened). So, for me I don't see any weak points." |

(*Continued*)

**Table 4.** (Continued)

| Mental Health Support | "For the mental health consultations, I found them to be very good, because when a woman is carrying a baby, above all, you have lot of hormones and you may sometimes have stress, you can have a problem in your house or with your environment. So, when we have a psychologist to talk to, then that reduces our stress and. . . for the environment, I managed my environment well and if I had a problem also that knocked me over, then we had the chance to talk with the psychologist to help us manage the problem. That had an effect on the baby, so it is an initiative that I loved so much and it is very good for us in J9." |
|---|---|

signs, she was able to explain to the staff exactly what was going on. She recognized that she could have lost her baby had she not known this information.

*Subtheme*: *Nutrition and Lifestyle changes*. Participants also expressed feeling empowered by the education about nutrition and lifestyle changes they received. The J9 team conducts nutritional education and demonstrations as part of the antenatal and pediatric group health literacy. One participant highlighted the importance of nutritional education on healthy balanced meals for themselves in pregnancy ". . .then when we eat these foods, it helps us have a healthy baby and us too, we are in good health, even after we deliver." She explains how she understood the different food groups and the role of the food groups to stay healthy and have a healthy baby.

*Subtheme*: *Information on care of the baby*. Educating parents on baby care starts prenatally, continuing through the group well-baby pediatric appointments. The team focuses on warning signs in the newborn, breastfeeding, weaning foods, and common infant illnesses parents should be aware of. One patient explains the importance of caring for and watching over their infants as was emphasized in the groups, in being aware of danger or warning signs. ". . . be sure to watch over the babies, to always be mindful of how we take care of the babies."

*Subtheme*: *Empowerment/ teaching others*. Interestingly, participants also reported feeling empowered to also teach others in the community based on what they had learned through J9 group care. One participant stated, "I have done a lot of trainings with other people" and referred two other people to J9 while expressing her love for J9. Pregnant women are referring one another into J9 is a testament to the satisfaction with integrated care. Other participants said they had heard about J9 from women who had been in the program; it has a reputation in the communities and many patients were referred to the program purely by word of mouth.

## Theme 2: Efficiency of the program

*"While you are learning they give you your prenatal check and you don't even take a lot of time"*

Participants highlighted how efficient they perceived the program compared to traditional prenatal care. Participants expressed that in usual care, "you spend almost the whole day". We developed two subthemes on components of the program the participants found efficient.

*Subtheme*: *Reliable and efficient antenatal care*. Participants expressed appreciation that their appointments were guaranteed and on time. As a participant stated, "Usually when you go for a prenatal visit, you spend almost the whole day, to get your chart, then they send you to get lab tests, sometimes you aren't even able to get seen. But with J9, it is different because there is a group and they call you sometimes, they call you for your appointment and when

you come, they have you sit down and while you are learning they give you your prenatal check and you don't even take a lot of time. If it is 8 o'clock you have an appointment, by 10 o'clock or before 10 o'clock you have left and are heading home. So as a pregnant woman sometimes you are not feeling well (have discomfort) so with the J9 program, it has everything you need; it is all that I need." Participants also felt that their 2-hour (group) appointments were efficient because not only did they receive health care but also benefitted from learning at the same time.

*Subtheme*: *Integrated care*: *Connection between antenatal care and hospital care*. Participants also appreciated the integration and coordination that existed between the J9 program with the hospital staff and facility. One patient's comment reinforced the interface and connection between various locations of care delivery: ". . .When I arrived in J9, they told me [about a problem] and they sent me to the Women's Emergency Room and there they measured me and checked everything, and they sent me back to J9 again." In this example of integrated care, the connection between antenatal and hospital care was reinforced through collaboration and communication between various providers. This patient explains not only the smooth referral to the Women's Health ER and back to outpatient J9, highlighting the direct impact of integrated programs on the patient.

## Theme 3: The love that is in J9

*"But the love that is inside J9, that impressed me even more"*

Another main theme was the respectful care that patients felt from their providers and other women in their group. We developed two subthemes that highlight this caring support.

*Subtheme*: *Personable and respectful care from providers*. J9 providers provide personable and respectful care for patients. Providers know them by name and call them to remind them about their appointments. One participant reiterated how the patients feel seen and heard and cared for: "I have found that they really take care of us. When I come in for a consultation they call me for my checkup, and they always welcome us very well." This patient explains perfectly what respectful care feels like for a vulnerable pregnant woman and the importance of feeling welcomed and truly cared for.

*Subtheme*: *Support from other patients*. Natural support groups and networks are formed among J9 patient groups during pregnancy and the babies' first year. This "family" that the patient formed in bonding with other mothers in her J9 group served to fill the need they had to truly be there for one another during their births: "That was like there were four sisters that were there, it was like we were shoulders for one another and when we delivered, we watched over the beds. When it was time to leave, they said I saw you deliver, I will wait for you. . . It is like we came to be a family. So, the love that is inside J9, I love that." This sisterhood support network, as this participant explains, allowed them to help one another during labor, delivery, and immediate post-partum, stating that they will wait for one another and later call on the phone, asking how they were doing. The patient expresses this deep caring as "love that is inside J9".

## Theme 4: Perception of reduced morbidity and mortality

*"Thanks to the J9 program everyone can notice that there are fewer children dying and fewer women who die from eclampsia as well"*

This last theme highlights the perception of fewer women and children dying in the community and the detection of warning signs both during facility visits and home visits; the counseling and support provided to vulnerable participants was highlighted in the three subthemes generated here.

*Subtheme*: *Prevention for mother*. This participant summarized the noticeable impact of J9 in terms of community perception of reduced morbidity and mortality. The participant explains that "everyone" is noticing that there are fewer women and children dying. The perception of averting avoidable deaths, she elucidates through her own personal experience with post-partum severe pre-eclampsia that the J9 program saved her life: "Because there is post-partum eclampsia, I could have had that because when they called me to come for the postpartum visit, I never knew that my blood pressure was so high. . . if they did not call me for my post-partum visit, I could have had eclampsia or I could have died." Here another aspect of the program is highlighted: post-natal care. The recognition and referrals to urgent care in cases with obstetrical warning signs pre- and post-natally is essential in reducing maternal mortality.

*Subtheme*: *Prevention for infant*. Another example of perception of reduced morbidity is highlighted when a participant explained the prompt recognition of pediatric warning signs during a home visit by the J9 team and the ease of community-to-hospital care. ". . .when they came to visit me, they saw the baby and they noticed that the umbilical cord had not fallen off yet. . . they saw that the umbilical area was not normal, and they said to me to urgently go to the hospital." The participant explains that during the home visit, the problem was identified during an examination by the J9 nurse accompagnateur. She was urged to take the infant immediately to the hospital for care. It is for reasons such as these that community-based care/home visits are a key pillar of the program.

*Subtheme*: *Mental health support*. From a programmatic angle, psychosocial support and counseling is another main pillar of J9. All antenatal and post-partum patients are screened for post-traumatic stress disorder and depression. This aspect of the program facilitates access to resources for patients who are victims of gender-based violence (GBV) or who have depression or other mental health needs. Some stress-reducing activities are also integrated in the group visits, an added benefit for the patients especially given the insecurity and socio-political instability in the country; additionally, ". . .If I had a problem also that knocked me over, then we had the chance to talk with the psychologist to help us manage the problem. That had an effect on the baby, so it is an initiative that I loved so much, and it is very good for us in J9." This quote highlights the impact that psychosocial support has on J9 patients in being able to cope with stressors.

## Discussion

We report the results of J9 Plus, a novel integrated accompaniment care model in maternal and child health care implemented in the Central Plateau of Haiti. In this cohort of 524 maternal-infant dyads receiving the J9 model of care and 523 receiving usual care, we did not detect differences in maternal and neonatal mortality, which remains high across the groups at 0.4–0.6% (40-60/10,000) and 1.2% (12/1000), respectively. The leading causes of maternal mortality in Haiti are late-stage severe pre-eclampsia and eclampsia (37.5%) and post-partum hemorrhage (22%), followed by infection and other causes [25], while the leading causes of neonatal mortality are complications of preterm birth, birth asphyxia, congenital anomalies, and infection [26]. Several important outcomes of the J9 model are therefore notable. The J9 model is associated with increased engagement in antenatal and postpartum/infant care, reduced rates of severe pre-eclampsia, and increased use of post-partum family planning methods, all factors

that may lead to lower rates of the common causes of morbidity and mortality in low- and middle-income countries (LMICs).

At Zanmi Lasante, the J9 model was associated with significantly higher attendance and engagement in antenatal and post-partum care compared to patients in usual care. Adherence to antenatal care was 70% vs. 34.2% for 4–7 visits and 23.5% vs. 1.9% for 8+ ANC visits respectively and 52% vs. 16% for postpartum care, respectively. As no transport or other incentive was given to participate, we hypothesize that the J9 model, based on the PIH comprehensive accompaniment model of community-focused care, is better meeting patients' needs and promotes patient confidence in their providers and therefore increased engagement with care. Overall, quantitative and qualitative findings of the study demonstrated that the J9 intervention was associated with similar outcomes of GANC studied in other LMICs [1–4] in terms of better patient education and information shared in the groups [1–3, 8], increased adherence to care [1–3], and increased family planning uptake [4].

As noted, our study documented a lower rate of pre-eclampsia with severe features in the J9 group compared to women receiving usual care. We hypothesize that J9 patients with elevated blood pressures were identified earlier during group antenatal care and home visits and that patients in J9 self-presented earlier to the hospital when they had warning signs, as a result of repetitive and reinforced health education during group visits and home visits. This was also supported by the themes identified in our qualitative data analysis, related to education, access to integrated care, and perceived efficacy of the model; "*. . .everyone can notice that there are fewer children dying and fewer women who die from eclampsia as well.*" The community perception of reduced morbidity and mortality is an important perspective when considering the uptake and sustainability of the J9 program.

We hypothesize that interventions to avoid progression of hypertensive disease of pregnancy and pre-eclampsia also impacted timing of delivery and mode of delivery. The J9 group had higher preterm birth and cesarean delivery rates and lower neonatal birthweights compared to the usual care group. The greater numbers of antenatal care visits among J9 participants afforded opportunities for increased screening for complications of pregnancy as well as more efficient referral for expedited delivery. Earlier presentation to care and expedited delivery via cesarean may also have contributed to the lower Apgar scores observed among neonates in the J9 group; these findings merit further study to elucidate the reasons for preterm birth and cesarean delivery among participants in the J9 model.

Accompaniment provides direct care to patients and establishes a close rapport while providing holistic support over the long term. Community-based care is a central component to this model of care. Providers, residents, and staff in outpatient and inpatient maternal-child health services have accompanied the J9 team on home visits to communities to provide care, supporting the social medicine aspect of the PIH approach to meeting patients' individual needs, addressing social determinants of health. This approach impressed upon our staff and trainees how these social determinants of health may impact patient outcomes and helped to strengthen the community's confidence in the health care team. From reports in 2019, 76% of patients enrolled in J9 received at least one home visit by the J9 team. We believe that this approach positively impacted our outcomes as the team had individual time with pregnant mothers and babies in their homes, permitting nutrition evaluation and counseling, water, sanitation, and hygiene (WASH), mother-child clinical evaluation, and couples' psychosocial evaluation and counseling, in addition to increasing access to family planning counseling and LARC subcutaneous implants placed in the privacy of patients' homes. Multi-generational counseling and education on key J9 messages exposed families to the same information that mothers receive in J9, which we theorize helped reinforce early presentation to care in cases of emergencies.

There is a high prevalence of perinatal anxiety and depression in LMICs, with depression affecting one in four women [27–30]. Perinatal anxiety and depression increases the risk for adverse health outcomes for mothers and newborns [31]. Psychological interventions, such as psychosocial support and counseling, have been shown to reduce perinatal anxiety and stress [32], which can impact perinatal outcomes [32]. As one of the main pillars of the program, patients in J9 have benefited from antenatal detection and treatment of antenatal and postpartum PTSD and depression. Our qualitative results show that patients valued the importance of the support that the psychosocial team provides: ". . .stress, you can have a problem in your house or with your environment, so when we have a psychologist to talk to, then that reduces our stress. . ." Additional research is needed to elucidate the impact of perinatal psychosocial support and counseling on maternal and newborn outcomes in J9.

As described, the J9 Plus maternal- child accompaniment model is translatable to other LMICs. Traditional antenatal care requires multiple providers to engage in numerous short one-on-one visits with little time for education and psychosocial support. It also requires multiple patient examination rooms and often results in long wait times and limited numbers of appointments leaving many patients with limited access to care. Moving to the J9 model of antenatal care required one larger space to meet in the group format (in our case 10–11 patients in a group), assignment of three personnel (a health care practitioner, a nurse, and an assistant) to assist with intake, essential screenings (e.g., blood pressure, fundal height, and fetal heart tone assessment), and record-keeping, followed by the interactive session focused on education and psychosocial support. It also required trainings of all team members and a commitment to interdisciplinary, educational, and patient-centered care. Additionally, the J9 Plus model included a psychologist and a social worker. Providing community-based care requires additional resources for travel and available personnel to leave the primary site of care on a weekly basis. We acknowledge that translating this model as described requires resources that all sites may not have readily available. However, the benefits of the model include more efficient and cost-effective means of providing high-quality care as noted in the GANC component of our model.

The benefits of an intervention such as J9 Plus are not limited to low resource countries like Haiti. The United States, a high-income country, reports a higher incidence of maternal morbidity and mortality among Black women compared to their white counterparts [33]. In studies in the US, GANC has been associated with lower rates of preterm birth and higher rates of breastfeeding compared to usual care [10, 34]. New models of care involving multidisciplinary approaches, attention to mental health, and community-based care are being proposed [35]. Improved perinatal and postpartum support inclusive of doulas has also been endorsed [36].

## Limitations

Our analysis has limitations that should be acknowledged in considering the results and planning future studies. First, the study cohort is limited to patients who delivered at HUM, as it was not possible for the team to gather maternal and neonatal outcome data on patients who delivered at home or in another facility. Likewise, the usual care/comparison group is limited to patients who had at least two documented antenatal care visits at HUM; those who had less than two visits were not included in the comparison group because there was limited access to their antenatal records and care, which may have occurred elsewhere or not at all. The exclusion of patients with minimal documented antenatal visits in the comparison group likely underestimated the measurable positive benefits of the J9 intervention compared to the general population because the comparison (usual care) group in this cohort likely had better care

than the general population where many women have little or no care and often give birth outside of the hospital.

Patients included in the J9 group voluntarily enrolled in this program and may reflect a group more predisposed to engagement with healthcare and its benefits. However, it is of note that enrollment was open to anyone desiring participation, without limitations or cap as long as they met inclusion criteria. Patients who started antenatal care late in pregnancy (after 20 weeks of gestation) could not enroll and often became members of the usual care group. These factors could bias results in favor of the J9 model of care.

Our data showed a significant difference in the ages between the two groups with lower proportion of teenagers present in the intervention versus usual care group (25 (4.8%) versus 58 (11.1%)) respectively. However, we controlled for maternal age in our analysis on maternal and neonatal outcomes. Future studies could focus on recruitment of a greater proportion of teenagers in the intervention group and evaluate its impact on outcomes.

Our study is subject to limitations associated with retrospective studies including misclassification and missing data for some variables. The post-partum hemorrhage rate of <1% in this cohort is not reflective of national statistics reporting a 22% rate of post-partum hemorrhage [25], but this rate is consistent with institutional statistics and may be reflective of how post-partum hemorrhage is documented at HUM, with only severe post-partum hemorrhage being recorded in the medical record. Likewise, intrauterine fetal demise and immediate perinatal demise were hard to differentiate as both were noted to have Apgar scores of zero at birth without additional documentation. Lastly, there is a lack of first trimester dating ultrasonography available for all patients at HUM. When last menstrual period was unknown or estimated, gestational age estimation was based on physical exam (fundal height) at antenatal visits. Therefore, this estimate could be a source of misclassification or bias if J9 participants had more accurate gestational age estimation due to the increase in number of antenatal visits.

The lack of robust electronic medical records maintained at facilities in the Central Plateau Department (region) of Haiti and the fact that the remaining data was collected from paper charts and entered on the CommCare app for data analysis led to missing and unavailable data for some variables. When this occurred, the number of subjects analyzed differed from the number in the total group. These differences varied between 0 and 16%. Over 10% missing data points were noted for: maternal parity in the usual care group (15%); low birth weight in the J9 group (12%); Apgar score in the J9 group (12%); gestational age in the usual care group (12%); and NICU admissions in the usual care group (14%).

In terms of the qualitative data gathered and analyzed, as explained above, only one focus group was coded and analyzed; however, we detected recurrent themes and sub-themes in the data provided by each of the participants and feel that the qualitative data analysis provides a holistic view of the patient experience with J9. Conducting individual and/ or focus groups with additional participants in the future would be important to determine if there are other themes regarding participant experiences with J9.

Lastly, this study was limited in its ability to compare infant outcomes in the first year of life among J9 participants and those in usual care. As the hospital is unable to provide standard "well-baby checks" or to prospectively follow infants in usual care, outcomes of pediatric group visits in J9, such as rates of malnutrition, hospitalization, and emergency visits, cannot at this time be meaningful compared to infants in usual care.

Fortunately, the J9 program continues to grow nationally and expand to other facilities providing an enlarging cohort in which to measure outcomes in the future. A larger programmatic evaluation to examine qualitative and quantitative outcomes on a greater scale would be beneficial.

## Conclusion

The J9 Plus maternal-child accompaniment model provides patient-centered and community-based holistic health care complemented by psychosocial support and education in antenatal and pediatric groups. Our study shows that in comparison to usual care, the J9 Plus model is associated with improved engagement with antenatal care and reduced rates of severe pre-eclampsia, the leading cause of maternal death in Haiti. In addition, the J9 model is associated with increased engagement with postpartum care and increased utilization of post-partum family planning methods. The accompaniment model is at the core of PIH's work, and this is the first time it has been studied in the perinatal health setting. While differences in maternal and neonatal mortality have not yet been demonstrated, the reduction in severe preeclampsia and the increased engagement with postnatal pediatric care are promising measured outcomes. As a model that is developed to meet the ongoing and emerging needs of patients and families in a community, the J9 approach can be implemented beyond Haiti in settings with poor or disparate maternal and neonatal outcomes impacted by social determinants of health and limited resources.

## Supporting information

**S1 File. Quantitative data.**
(XLS)

**S2 File. Qualitative data.**
(DOC)

## Acknowledgments

The team would like to thank Emmanuel Demosthene and Jimmy Jean-Baptiste for all their support and help with data collection.

## Author Contributions

**Conceptualization:** Meredith Casella Jean-Baptiste, Marc Julmisse, Oluwatosin O. Adeyemo, Thamar Monide Vital Julmiste, Jessica L. Illuzzi.

**Data curation:** Meredith Casella Jean-Baptiste.

**Formal analysis:** Meredith Casella Jean-Baptiste, Oluwatosin O. Adeyemo, Jessica L. Illuzzi.

**Investigation:** Meredith Casella Jean-Baptiste, Thamar Monide Vital Julmiste.

**Methodology:** Meredith Casella Jean-Baptiste, Marc Julmisse, Oluwatosin O. Adeyemo, Jessica L. Illuzzi.

**Project administration:** Meredith Casella Jean-Baptiste, Thamar Monide Vital Julmiste.

**Supervision:** Meredith Casella Jean-Baptiste, Thamar Monide Vital Julmiste.

**Writing – original draft:** Meredith Casella Jean-Baptiste, Marc Julmisse, Oluwatosin O. Adeyemo, Thamar Monide Vital Julmiste, Jessica L. Illuzzi.

**Writing – review & editing:** Meredith Casella Jean-Baptiste, Marc Julmisse, Oluwatosin O. Adeyemo, Jessica L. Illuzzi.

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
