## [Decision Letter · Decision Letter 0]

18 Apr 2024

PONE-D-24-08229Integrated group antenatal and pediatric care in Haiti: A comprehensive care accompaniment modelPLOS ONE

Dear Dr. Casella Jean-Baptiste,

Thank you for submitting your manuscript to PLOS ONE. After careful consideration, we feel that it has merit but does not fully meet PLOS ONE’s publication criteria as it currently stands. Therefore, we invite you to submit a revised version of the manuscript that addresses the points raised during the review process.

We look forward to receiving your revised manuscript.

Kind regards,

Ochuwa Adiketu Babah, FWACS, FMCOG

Academic Editor

PLOS ONE

Additional Editor Comments:

Dear Authors,

Thank you for submitting your manuscript to PLOS One. The review process has been completed. There are comments below for you to address, after wish a decision on the suitability of the manuscript for publication in PLOS One will be made.

Reviewer coments:

The authors sought to evaluate the impact of the J9 model of care on perinatal outcomes among women in Haiti by employing mixed methods.

Below are a few comments for authors to consider.

It seems there was a significant difference between the ages of cases and controls (p-0.001). Was matching done? If not, was it considered in the analysis as a confounder since this could impact on the outcome measure?

If the mean of ages was chosen as a measure of central tendency, then the standard deviation must be indicated and not the range. Did the data show a normal distribution? The authors should clarify. If not, the median (IQR) will be preferred to the mean.

Table 2. Authors should ensure all the variables with their respective totals are well tabulated including the responses under each category of the variable for consistency (e.g. Delivery type, preeclampsia/eclampsia, post-partum visit < 6 weeks, etc). This has been done in Table 3.

In the qualitative aspect of the study, the authors indicated only one FGD (with 8 participants) was used because of the inability to obtain consent from the other study participants involved in the other FGD undertaken. Was this challenge rather not a limitation in achieving saturation or did it have no impact on the outcome?

Language editing is required in the sentence at lines 437-439.

Best regards,

Reviewers' comments:

Reviewer's Responses to Questions

**Comments to the Author**

1. Is the manuscript technically sound, and do the data support the conclusions?

Reviewer #1: Yes

Reviewer #2: Yes

2. Has the statistical analysis been performed appropriately and rigorously? 

Reviewer #1: Yes

Reviewer #2: Yes

3. Have the authors made all data underlying the findings in their manuscript fully available?

Reviewer #1: Yes

Reviewer #2: No

4. Is the manuscript presented in an intelligible fashion and written in standard English?

Reviewer #1: Yes

Reviewer #2: Yes

5. Review Comments to the Author

Reviewer #1: Thank you for producing such a well dedicated and informative study ,i appreciated it especially regardind the methodology ,abstract ,i advise to continue further research to get more reproducible results.

Reviewer #2: The authors sought to evaluate the impact of the J9 model of care on perinatal outcomes among women in Haiti by employing mixed methods.

Below are a few comments for authors to consider.

It seems there was a significant difference between the ages of cases and controls (p-0.001). Was matching done? If not, was it considered in the analysis as a confounder since this could impact on the outcome measure?

If the mean of ages was chosen as a measure of central tendency, then the standard deviation must be indicated and not the range. Did the data show a normal distribution? The authors should clarify. If not, the median (IQR) will be preferred to the mean.

Table 2. Authors should ensure all the variables with their respective totals are well tabulated including the responses under each category of the variable for consistency (e.g. Delivery type, preeclampsia/eclampsia, post-partum visit < 6 weeks, etc). This has been done in Table 3.

In the qualitative aspect of the study, the authors indicated only one FGD (with 8 participants) was used because of the inability to obtain consent from the other study participants involved in the other FGD undertaken. Was this challenge rather not a limitation in achieving saturation or did it have no impact on the outcome?

Language editing is required in the sentence at lines 437-439,

6. PLOS authors have the option to publish the peer review history of their article (what does this mean?). If published, this will include your full peer review and any attached files.

Reviewer #1: No

Reviewer #2: No

---

## [Author Response · Author response to Decision Letter 0]

4 Jun 2024

RESPONSE TO REVIEWERS AND EDITOR

Journal requirements: (ACADEMIC EDITOR)

RESPONSE: Done

2. We note that you have indicated that there are restrictions to data sharing for this study. PLOS only allows data to be available upon request if there are legal or ethical restrictions on sharing data publicly. For more information on unacceptable data access restrictions, please see http://journals.plos.org/plosone/s/data-availability#loc-unacceptable-data-access-restrictions .

RESPONSE: Done. We received approval to share the data.

RESPONSE: Done

Reviewer comments:

The authors sought to evaluate the impact of the J9 model of care on perinatal outcomes among women in Haiti by employing mixed methods.

Below are a few comments for authors to consider.

1. It seems there was a significant difference between the ages of cases and controls (p-0.001). Was matching done? If not, was it considered in the analysis as a confounder since this could impact on the outcome measure?

RESPONSE: Thank you for your comment. Matching was not done as our study was not a randomized control trial. However, we do have a lower proportion of teenagers in the intervention group. Hence, in response to your comments, we controlled for maternal age in our logistic regression . We have updated the results and limitation sections to reflect this.

2. If the mean of ages was chosen as a measure of central tendency, then the standard deviation must be indicated and not the range. Did the data show a normal distribution? The authors should clarify. If not, the median (IQR) will be preferred to the mean.

RESPONSE: Thank you for this comment. The age data were normally distributed, both the Skewness and Kurtosis are between -2 and 2, [Skewness for J9 is 0.15 and Kurtosis is –0.49 and for the Usual care group Skewness is 0.25 and Kurtosis is –0.63]. In response to your comment, we added a statement to the result section stating that the maternal age was normally distributed. We also added the standard deviation into Table 1.

3. Table 2. Authors should ensure all the variables with their respective totals are well tabulated including the responses under each category of the variable for consistency (e.g. Delivery type, preeclampsia/eclampsia, post-partum visit < 6 weeks, etc). This has been done in Table 3.

RESPONSE: This has been corrected for all tables. 

4. In the qualitative aspect of the study, the authors indicated only one FGD (with 8 participants) was used because of the inability to obtain consent from the other study participants involved in the other FGD undertaken. Was this challenge rather not a limitation in achieving saturation or did it have no impact on the outcome?

RESPONSE; Thank you for this comment, we agree with this limitation We revised our limitations section to further address this in the 7th paragraph. 

“In terms of the qualitative data gathered and analyzed, as explained above, only one focus group was coded and analyzed; however, we detected saturation of recurrent themes and sub-themes in the data provided by each of the participants and feel that the qualitative data analysis provides a holistic view of the patient experience with J9. Conducting individual and/ or focus groups with additional participants in the future would be important to determine if there are other themes regarding participant experiences with J9.”

5. Language editing is required in the sentence at lines 437-439.

RESPONSE: Thank you for this comment. We changed the wording to make it less confusing to: “There is a high prevalence of perinatal anxiety and depression in LMICs with depression affecting one in four women [26,27,28,29]. Perinatal anxiety and depression increases the risk for adverse health outcomes for mother and newborn [30].”

---

## [Editor Report · Decision Letter 1]

18 Jun 2024

Integrated group antenatal and pediatric care in Haiti: A comprehensive care accompaniment model

PONE-D-24-08229R1

Dear Meredith Casella Jean-Baptiste,

We’re pleased to inform you that your manuscript has been judged scientifically suitable for publication and will be formally accepted for publication once it meets all outstanding technical requirements.

Kind regards,

Ochuwa Adiketu Babah, FWACS, FMCOG

Academic Editor

PLOS ONE

---

## [Editor Report · Acceptance letter]

4 Jul 2024

PONE-D-24-08229R1 

PLOS ONE

Dear Dr. Casella Jean-Baptiste, 

I'm pleased to inform you that your manuscript has been deemed suitable for publication in PLOS ONE. Congratulations! Your manuscript is now being handed over to our production team.

Kind regards, 

on behalf of

Dr. Ochuwa Adiketu Babah 

Academic Editor

PLOS ONE